# Individualized Private Graph Neural Network via Node Influence-based Noise Adaptation

## Abstract

Graph Neural Networks (GNNs) with Differential Privacy (DP) guarantees have been proposed to preserve privacy when nodes contain sensitive information that needs to be kept private but is critical for training. Existing methods deploy a fixed uniform noise generation mechanism that lacks the flexibility to adjust between nodes, leading to increasing the risk of graph information leakage and decreasing the model's overall performance. To address the above challenges, we propose NIP-GNN, a Node-level Individual Private GNN with DP guarantee based on the adaptive perturbation over sensitive components to safeguard node information. First, we propose a Topology-based Node Influence Estimation (TNIE) method to infer unknown node influence with neighborhood and centrality awareness. Second, given the obtained node influence rank, an adaptive private aggregation method is proposed to perturb neighborhood embeddings directed by node-wise influence. Third, we propose to privately train the graph learning algorithm over perturbed aggregations in adaptive residual connection mode over multi-layer convolution for node-wise tasks. Theoretically, analysis ensures that NIP-GNN satisfies DP guarantee. Empirical experiments over real-world graph datasets show that NIP-GNN presents a better resistance over node inference attacks and achieves a better trade-off between privacy and accuracy.

## 1 Introduction

In recent years, Graph Neural Networks (GNNs) have achieved outstanding performance in several domains, such as social analysis (Yang et al., 2021b), financial anomaly detection (Chen et al., 2020), time series analysis (Wang et al., 2021), and molecule synthesis (Gasteiger et al., 2021). Through aggregating the feature of neighboring nodes and fully mining and fusing the topological associations in graph, GNNs yield state-of-art performance in tasks such as link prediction (Zhao et al., 2021), node classification (Guan et al., 2022), and sub-graph classification (Yang et al., 2021a). However, graph data in the real world usually contain private information (Li et al., 2023). For example, in social network graphs where nodes denote users, and edges indicate the existence of social attributes like being friends. Node features carry sensitive information, such as the average online time of users per week.

**Necessity and motivation of private GNN.** Directly training over the graph contains sensitive information may lead to unignorable privacy leakage. Attackers have the ability to infer the existence of arbitrary node, attribute or link, or specific graph-level statistic information like average degree, by accessing GNN immediately (like GNN embedding) or final result (like GNN output), as shown in membership inference attack (Wang & Wang, 2022; Zhang et al., 2022b), attribute inference attack Olatunji et al. (2023a), edge stealing attack (Wu et al., 2022), and graph reconstruction attack (Zhang et al., 2022b). This raises the necessity of protecting graph data privacy in GNN.

Among all the privacy-preserving technologies, Differential Privacy (DP) (Dwork et al., 2014) emerges to be a widely used method for its advantage of strict theoretical guarantee (Zhang et al., 2021) and flexible control of protection strength by adjusting privacy budget (Sala et al., 2011). Existing graph neural networks under DP guarantee can be divided into *edge-level* (Wu et al., 2022; Zhu et al., 2023) and *node-level* (Daigavane et al., 2021; Sajadmanesh et al., 2022; Sajadmanesh & Gatica-Perez, 2023) protection, where the former add the non-trivial calibrated noise into edges to prevent adversary from inferring edge existence (Wu et al., 2022; Zhu et al., 2023), and the latter

add calibrated noise into loss gradients (Daigavane et al., 2021) or message-passing aggregations (Sajadmanesh et al., 2022; Sajadmanesh & Gatica-Perez, 2023) to prevent adversary from inferring node existence, including node feature, edges and label. We focus on the *node-level DP on graph* in this paper considering it protects the more comprehensive information and is more challenging.

However, existing GNNs under DP guarantee deploy a uniform noise generation mechanism that lacks the flexibility to adjust between nodes, leading to two problems: First, difficulty in adequately protecting high-influence nodes, increasing the risk of graph information leakage. Second, the inability to meet the diverse privacy needs among nodes makes it hard to balance the trade-off between graph privacy and model utility, ultimately affecting the model's overall performance.

**Technical Challenges of Individualized Private GNN under DP.** To the best of our knowledge, no existing fine-grained DP technologies specifically address the flexible privacy needs among nodes. It is non-trivial to satisfy the desired protection level from both the theoretical and practical aspects for GNNs with DP constraints. We contend that individualized differentially private GNN poses three significant challenges, rendering other private learning over structure and graph data. **(1). Typical DP Stochastic Gradient Descent (DPSGD) based privacy technology is not directly suitable for GNN.** As a widely used technology, DPSGD (Abadi et al., 2016) adds noise over the clipped gradient of data in the randomly selected batch. Notice that the calibrated noise is proportion to query sensitivity over data, where sensitivity measures the largest impact of arbitrary sample (Dwork et al., 2014). However, due to the complexity of topology links, the node-level query sensitivity of GNNs is high since each node influences its neighbors in message-passing and aggregation. When applying DPSGD with GNN optimization, sensitivity escalates from the clipped norm to the batch size. **(2). Complexity of individualized noise calibration and injection.** Nodes' various roles and influence (Scripps et al., 2007; Lawyer, 2015) in the graph complicate the individualized noise calibration. The proposed fine-grained privacy assessment and noise generation scheme needs to consider features such as node degree, and centrality. **(3). Optimize the negative effect of injected noise on private GNN utility.** Private GNNs inevitably sacrifice utility to ensure model privacy (Sajadmanesh & Gatica-Perez, 2023), which poses the effort to better balance the trade-off between GNN utility and graph data privacy. Fine-grained noise injection leads to differentiated noise accumulating and propagating through multiple layers and neighbor aggregations. Therefore, gradually mitigating the noise's negative impact on model utility during the neighbor message-passing layers is a key direction.

**Our Solutions and Contributions.** To address these challenges, we propose a Node-level Individual Private Graph Neural Network (NIP-GNN) with DP guarantee, which flexibly adjusts node protection level based on learnable influence and independent of the training epoch. Our goal is to develop a fine-grained and adaptive differentially private GNN that distributes a more granular privacy budget and achieves a better trade-off between utility and privacy. First, Topology-based Node Impact Estimation (TNIE) method is proposed to capture node influence with neighborhood and centrality awareness adjustments. Second, we propose an influence-awareness fine-grained permutation method to realize node-diverse privacy-preserving distribution in GNN by injecting diversity-calibrated noise into immediate aggregations. Third, an adaptive message-passing layer with residual selective cooperation is proposed to improve model utility by optimizing the negative effect of previously injected noise. We show theoretically that NIP-GNN satisfies the DP guarantee. Experiments on real graph datasets show that the proposed method achieves a better trade-off between the utility-privacy, compared with existing private GNNs. The main contributions are summarized as:

- We study a new solution of differentially private GNN, Node-level Individual Private Graph Neural Network (NIP-GNN), with flexible privacy calibration and adaptive aggregation layer to achieve better utility. To the best of our knowledge, this is the first work to focus on the fine-grained DP mechanism for GNN.

- We propose the Topology-based Node Influence Estimation (TNIE) method to infer nodes' impact over local and global structures directly by neighborhood and centrality awareness. Differentially private graph embedding is obtained by calibrating fine-grained noise among nodes, which is determined by influence rank, and injecting it into immediate embeddings. The adaptive layer is proposed to improve utility by mitigating noise's impact during neighbor message-passing and update by selective optimization.

- Theoretical analysis proves that NIP-GNN satisfies DP requirement. The experimental results on four benchmark graph datasets demonstrate the prior utility of NIP-GNN.

## 2 RELATED WORKS

Recently, there have been several attempts to use DP to provide node-level and edge-level privacy guarantees in GNNs. For node-level protection, existing methods can be divided into two types: gradient-based and aggregation-based perturbation. For gradient-based methods, Daigavane *et al.* Daigavane et al. (2021) proposed a one-layer node-level private GNN by extending the DPSGD algorithm to a degree-bounded graph. But Daigavane et al. (2021) is limited to one-layer GNN and can not preserve privacy for high-layer aggregations. Zhang et al. (2022a) proposes to convert the aggregation over the edge into aggregation over approximate personalized PageRank vectors to achieve edge-level protection. The node-level protection in the training process is achieved with DPSGD. For aggregation-based perturbation methods, Sajadmanesh et al. (2022) provides a private GNN by injecting uniform noise on the neighbor aggregation vectors. ProGAP proposed in Sajadmanesh & Gatica-Perez (2023) proposes to split the GNN training process into overlapping sub-models to achieve node-level and edge-level protection. The other edge-level protection methods over GNN, privacy attacks in GNNs, and existing privacy budget allocation methods over traditional deep learning are shown in Appendix A.1.

Despite the presence of uniform noise injection methods in differentially private GNNs and fine-grained noise allocation schemes in differentially private DL, such considerations are still missing in differentially private GNNs. The graph structure poses new challenges of noise complex calibration to DP technologies, existing fine-grained DP methods can not be directly used in GNN.

## 3 PROBLEM FORMULATION

In this section, we first revisit the definition of GNN and DP. Then we define the problem of private learning GNN with node privacy concerns.

### 3.1 GRAPH NEURAL NETWORK

Let $G = \{\mathcal{V}, \mathcal{E}\}$ be an unweighted undirected graph, where $\mathcal{V}$ and $\mathcal{E}$ denote the nodes set and edges set. The adjacency matrix $\mathbf{A} \in \{0, 1\}^{N \times N}$ represents the link among edges, $|N|$ denotes the node number. For $\forall v_i, v_j \in \mathcal{V}$, if there exists an edge between $v_i, v_j$, then $\mathbf{A}_{ij} = 1$, for else $\mathbf{A}_{ij} = 0$. Node feature of $v_i$ is a $d$-dimension vector, and the N $\times$ d matrix $\mathbf{X}$ represents the stack of all nodes' feature, where $\mathbf{X}_i \in \mathbf{X}$ denotes the feature of $v_i$. $\mathbf{Y} \in \{0, 1\}^{N \times M}$ represents the label of nodes, $\mathbf{Y}_i$ is a M-dimension one-hot vector, where M is the class number.

The typical message-passing-based GNN consists of two phases: message aggregation and updating. In the message aggregation phase of $i$-th layer, every node shares and receives neighbors embedding of the former $i$-1-th layer and outputs a new embedding after applying a transformation, which can be defined as:

$$E_j^i = f_{agg}(\{h_u^{i-1}, u \in \mathcal{N}(v_j)\}), \quad (1)$$

where $\mathcal{N}(v_j)$ denotes the adjacent node set of node $v_j$, and $h_u^{i-1}$ represents the embedding output of node $u$ at $i$-1-th layer. $f_{agg}$ is the aggregation linear function like SUM, MEAN, MAX, etc. $E_j^i$ is the aggregate output of node $v_j$ in $i$-th layer after the aggregation transformation of all adjacent nodes. Update transformation is employed on the $E_j^i$, which is shown as:

$$h_j^i = f_{upd}(E_j^i, h_j^{i-1}; \theta_j), \quad (2)$$

where $f_{upd}$ denotes the learnable function that takes the aggregate vector $E_j^i$ and last layers' embedding $h_j^{i-1}$ as input, and outputs the updated embedding of $v_j$ at $i$-th layer. $f_{upd}$ is determined by parameter $\theta_j$. The input $h_j^0$ of GNN's first layer is $\mathbf{X}_j$, and the last layer generates embedding vectors $h_j^L$, which can be used in downstream tasks. $L$ represents the total layer. A softmax layer is employed on the final embedding vectors $h_v^L$ to get the class probability of $v_j$. Following (Sajadmanesh et al., 2022; Chien et al., 2023), we focus on the node classification task.

### 3.2 PROBLEM DEFINITION

The goal of this paper is to preserve the adaptive privacy of the graph nodes, ensuring that the training and inference of GNN by following DP constraint, which quantifies the privacy-preserving

level by setting a privacy budget to measure the attack success probability. Different from previous work in (Daigavane et al., 2021), we aim to propose an epoch-independent method considering node influence from topology without losing much utility of GNN. We first define the notion of a Node-level adjacent graph as follows:

**Definition 1** (Node-level adjacent graph (Sajadmanesh et al., 2022)). *Graphs $G$ and $G'$ are node-level adjacent graphs if at most one node is different, including node features, links, and labels. Without loss of generality, let $G$ can be obtained by altering a node in $G'$.*

Then the $\epsilon$-Node-level differential privacy is defined as:

**Definition 2** ($\epsilon$-Node-level differential privacy (Sajadmanesh et al., 2022)). *Let $G$ and $G'$ be two node-level adjacent graph, given $\epsilon > 0$, the random algorithm $\mathcal{A}$ is $\epsilon$-Node-level differential privacy if for any set of outputs $S \in Range(\mathcal{A})$, satisfies:*

$$Pr[\mathcal{A}(G) \in S] \leq e^\epsilon \, Pr[\mathcal{A}(G') \in S]. \tag{3}$$

Here, $\epsilon$ is called the privacy budget, which is used to measure the protection extent. A higher $\epsilon$ means a higher protection level and more noise injection is needed.

Based on Definition 2, the global graph sensitivity can be defined as:

**Definition 3** (Graph $L_1$ sensitivity). *The global graph $L_1$ sensitivity of function $f$ on two node-level adjacent graphs $G$ and $G'$ is:*

$$\Delta_{gG} = max||f(G) - f(G')||_1 \tag{4}$$

DP has the following classic properties which support us in building complex algorithms over graph:

**Theorem 1** (Post-processing (Dwork & Lei, 2009)). *Post-processing to any $\epsilon$-DP algorithm's output remains $(\epsilon)$-DP.*

**Theorem 2** (Sequential composition (Dwork & Lei, 2009)). *If an $\epsilon_2$-DP algorithm is applied to $\epsilon_1$-DP algorithm's output, then the result is at most $(\epsilon_1 + \epsilon_2)$-DP.*

Based on the above definition, the problem of private GNN under DP constraints can be defined as:

**Definition 4** ($\epsilon$-Node-level Differentially Private Graph Neural Network). *Give a graph $G$ with nodes containing sensitive information, a well-trained GNN model $\mathcal{F}$ is a $\epsilon$-Node-level Differentially Private Graph Neural Network if for any Node-level adjacent graph $G'$ of $G$ and any outputs $S$ of $\mathcal{F}$, we have:*

$$Pr[\mathcal{F}(G) \in S] \leq e^\epsilon \, Pr[\mathcal{F}(G') \in S] \tag{5}$$

Therefore, the key of this paper is to propose a specific GNN model and design DP mechanisms, then consider how to incorporate the proposed DP mechanisms into the GNN training and inference phase to protect the training graph data from being theft, while keeping the private GNN model utility to satisfy downstream task requirements.

**Remark 1.** *We focus on the transductive learning in this paper for it considers more challenging privacy risk in the inference phase (Sajadmanesh et al., 2022), where the test nodes can still access train nodes features. The proposed method is suitable for inductive learning.*

## 4 PROPOSED METHOD: NIP-GNN

We propose our NIP-GNN composing *Topology-based Node Influence Estimation* (TNIE), *Node-Influence-Grained Adaptive Permutation* (NAP) and *Adaptive Calibrated Aggregation* (ACA). TNIE measures node influence in the graph directed by topology-related and feature-related awareness. NAP proposes to calibrate adaptive noise tailored for estimated node influence sequence constraint and permutes the propagation layer by adding generated node-wise adaptive noise. The analysis shows that NAP satisfies the DP definition. ACA proposes an adaptive aggregation layer with residual connection to balance and smooth the protective noise introduced by NAP. We also present an analysis on the total privacy level of the whole process.

### 4.1 Topology-based Node Importance Estimation (TNIE)

To estimate node influence in a given graph, considering that the known information of a specific node is its original feature and edges, we formulate the evaluation from two aspects: feature-related and topology-related mining. For feature-related awareness mining, TNIE maps the original node features to an immediate space to extract influence-driven embedding by a score computing function:

$$h_u^0 = \text{Score Computing}(\mathbf{X}_j\mathbf{u}; v_u \in \mathcal{V}), \tag{6}$$

where $v_u$ is a node in $G$. The score computing function takes node's original feature as input and output encoded embedding. Considering Multi-Layer Perception (MLP) is a general and foundational neural network, we use it in the paper. Other complex neural networks can also be used.

For topology-related awareness mining, we propose neighborhood and centrality sensing. Neighborhood sensing is based on the intuition that a node and its neighbors have mutual influence, so the neighbors' impact serves as an effective proxy for the node's significance. Centrality sensing suggests that nodes with higher centrality exert more influence than those with lower centrality, as they propagate messages to more nodes. For neighborhood sensing, TNIE builds a weighted aggregation from node $v_u$ and its neighbors for the $t$-th layer ($t = 1, 2, ...T$) to generate normalized representation:

$$h_u^t = \sum_{v \in N(u)} \frac{1}{\sqrt{D_u}} \frac{1}{\sqrt{D_v}} h_v^{t-1}, \tag{7}$$

where $D_u$ denotes degree of $u$. After $T$ layers aggregation, $\forall u \in V$ gets neighborhood sensing embedding $h_u^T$. For centrality sensing, considering node degree is commonly used as a proxy for centrality, we construct a centrality-driven embedding by integrating an adjustable centrality metric with the neighborhood sensing embedding. It seems natural to directly use the transformation of degree in fusion, however, the absolute value of the degree does not always accurately represent the nodes influence rank (Liao et al., 2017; Ibnoulouafi et al., 2018). Therefore, instead of initial degree $log(D_u)$ of node $v_u$, TNIE adopts a shifting degree $\lambda(log(D_u)) + \phi$ to allow the possible discrepancy between degree and influence rank, where $\lambda$ and $\phi$ are learnable parameters. Then the shifting degree is used to adjust the neighborhood awareness-based embedding $h_u^T$ of node $v_u$ from centrality sensing consideration, which is as follows:

$$s(u)^* = \sigma(\lambda(log(D_u) + \phi) \cdot h_u^T), \tag{8}$$

where $\sigma$ is a non-linear activation function, $s(u)^*$ denotes estimated influence score.

### 4.2 Node-Influence-Grained Adaptive Permutation (NAP)

NAP aims to privately generate and release nodes aggregation embedding by perturbing embedding via adaptive noise proportion to sensitivity and calculated node influence. Motivated by the fact that perturbing a node $v_u$'s edges in the graph can be seen as changing neighborhood aggregation of $v_u$'s adjacent nodes $\forall v \in N(u)$, we propose to calibrate noise generated by Laplace Mechanism (Definition 5) and inject it on the first layer aggregation embedding. In particular, we use the sum aggregation function as the first layer, which is equivalent to the multiplication of the adjacent matrix and the input row-normalized feature. The permutation process can be presented as follows:

$$\overline{\mathcal{H}^0}(\mathbf{A}, \mathcal{V}, \mathbf{X}) = \{\overline{\mathbf{H}_u^0}\}_{v_u \in \mathcal{V}} \ s.t. \ \overline{\mathbf{H}_u^0} = \sum_{j=1}^{|N|} \mathbf{A}_{uj}\mathbf{X}_j j + Lap(\frac{\Delta\mathcal{H}^0}{\epsilon_u}), \tag{9}$$

where $\mathbf{H}_u^0 = \sum_{j=1}^{|N|} \mathbf{A}_{uj}\mathbf{X}_j j$ denotes the sum aggregation process of node $v_u$, $\mathbf{A}_{uj} \in \mathbf{A}$, $\mathbf{X}_j j$ is the row-normalized feature of node $v_j$, $Lap(\frac{\Delta\mathcal{H}^0}{\epsilon_u})$ is the perturbed noise, $\Delta\mathcal{H}^0$ denotes the sensitivity of aggregation function, $\epsilon_u$ is the privacy budget of $v_u$. Note that our designed NAP is based on the widely-used Laplace Mechanism:

**Definition 5** (Laplace Mechanism (Dwork & Lei, 2009)). *Given an algorithm $\mathcal{A} \to \mathcal{D}^d$, the Laplace mechanism outputs $\mathcal{M}(G) = \mathcal{A}(G) + \gamma$, where $\gamma \sim Lap(\alpha)^d$ and $Lap(\alpha)^d$ is a length of $d$ vector samples from a Laplace distribution with scale $\alpha$. If $\alpha = \frac{\Delta_{gG}}{\epsilon}$, then the Laplace mechanism satisfies $\epsilon$-Node-level differential privacy.*

**Lemma 1.** *Let $G = \{\mathcal{V}, \mathcal{E}\}$ and $G' = \{\mathcal{V}', \mathcal{E}'\}$ be two adjacent graph. The global $L_1$ graph sensitivity of first sum aggregation layer $\Delta \mathcal{H}^0 \leq 2D_{max}$, where $D_{max}$ is the maximum node degree.*

Proof of Lemma 1 is shown in Appendix A.2.1. Denote the estimated influence rank of node $v_u$ as $\mathcal{R}\_\mathcal{NI}(u)$, which is gained through the rank of estimated score $s(u)*$. $\mathcal{C}(u)$ represents the privacy protection level of node $v_u$. For node $v_m$ and $v_n$, if $s(m)* > s(n)*$, we aim to realize $\mathcal{C}(m) > \mathcal{C}(n)$. Then let $\epsilon_A$ denotes the total privacy cost of first aggregation layer, the assigned privacy budget $\epsilon_u$ of $v_u$ is proportional to $\mathcal{C}(u)$, which is $\epsilon_u = \epsilon_A \cdot \beta_u$, where $\beta_u$ is a weight coefficient. From definition 5 and lemma 1, we know that injected noise is proportional to the maximal node degree. However, many real-world graphs follow power-law distribution (Clauset et al., 2009). When node degree distribution is extremely imbalanced, the maximum node degree is obviously higher than most nodes, Laplace-based noise generation mechanism may yield high noise. To tackle this challenge, we propose to leverage the potential wasted privacy budget generated by the node degree gap of adjacent graph nodes.

For arbitrary node $v_u \in V$, it receives a potential reusable privacy budget from neighboring node $v_k \in N(u)$. Total reusable privacy ratio of $v_k$ is $D_{max} - D_k$, for each neighbor node of $v_k$, the assigned ratio $r(u, k)$ is

$$r(u, k) = \frac{\mathcal{R}\_\mathcal{NI}(u)}{\sum_{v_j \in N(k)} \mathcal{R}\_\mathcal{NI}(j)}, \tag{10}$$

which is in proportion to node influence rank. Then the weight coefficient of $v_u$ is the minimum of all reusable budget ratios:

$$\beta_u = \min_{v_k \in N(u)} \left\{ \frac{\mathcal{R}\_\mathcal{NI}(u)}{\sum_{v_j \in N(k)} \mathcal{R}\_\mathcal{NI}(j)} (D_{max} - D_k) + 1, \frac{D_{max}}{D_k} \right\}. \tag{11}$$

Then based on node-wise privacy budget computed via equ.(11), differentially private aggregation is obtained by equ.(9). The generated private embedding is unbiased. Theoretical analysis of the following theorem and proposition is shown in Appendix A.2.1.

**Theorem 3.** *Algorithm 1 preserves $\epsilon_A$-DP in the first differential private aggregation layer $\mathcal{H}^0$.*

**Proposition 1.** *The sum aggregator defined in (9) for the first layer is unbiased.*

## 4.3 ADAPTIVE CALIBRATED AGGREGATION (ACA)

Laplace noise introduced by previous modules inevitably affects GNN performance. To achieve a better privacy-utility trade-off, we propose ACA, which adaptively aggregates noisy embeddings using residual connections across layers. ACA smooths the noise by iteratively and selectively aligning noisy neighbor aggregations with the node's own private sum embedding, based on the intuition that consistent information flow can mitigate noise impact while preserving essential features. ACA takes the $\overline{\mathcal{H}^0}$ as input and outputs the final embeddings Compared with equally aggregating neighbors embedding, ACA allows each node to learn from different neighbors with different weights.

To keep edges and node labels private, we perturb them before the further message passing and updating steps. For edges perturbation, we propose a degree-preserving edge randomization method, which reduces the impact of adding or removing edges on the graph by unbiasedly sampling the edges before and after edge randomization. First initializes an all-zero matrix $\tilde{\mathbf{A}}$ as the output matrix. Then, for each edges $(v_i, v_j)$, samples a value $x \sim \text{Bern}(1 - s)$ using the privacy parameter $s$ to decide whether to preserve the original edge. If the sampled result is $x = 1$, the original edge is preserved, setting $\tilde{\mathbf{A}}_{ij} = \mathbf{A}_{ij}$ and $\tilde{\mathbf{A}}_{ji} = \mathbf{A}_{ij}$. Otherwise, a value $y \sim \text{Bern}(1/2)$ is drawn from the Bernoulli distribution, and both $\tilde{\mathbf{A}}_{ij}$ and $\tilde{\mathbf{A}}_{ji}$ are set to $y$, effectively randomizing the edge's state. To reduce the impact of adding or removing edges on the graph, an unbiased sample method is proposed at the end with the sampling probability of:

$$p_u^{sample} = \frac{2D_u}{D_u + N - Ns + D_u s} \tag{12}$$

for node $v_u$, where $D_u$ is the degree of $v_u$ before permutation, $s$ is a parameter satisfying $s \geq \frac{2}{e^{\epsilon_B/2D_{max}}+1}$, $\epsilon_B$ is the privacy budget (the constraint of parameter $s$ in shown in Theorem 4). The

expectation of sampled node degree $\mathbb{E}(\overline{D'_u})$ equals $\mathbb{E}(D_u)$ for $\forall v_u \in \mathcal{V}$. The final noisy adjacent matrix is the unbiased $\overline{\mathbf{A}}$. The proposed sampled matrix perturbation satisfies DP guarantee.

**Theorem 4.** *Let $D_u$ denote the degree of $v_u$ in the original matrix $\mathbf{A}$, $\overline{D'_u}$ denotes the degree after perturbation and sampling. Then $\mathbb{E}(\overline{D'_u}) = \mathbb{E}(D_u)$. When the privacy parameter $s$ satisfies $s \geq \frac{2}{e^{\epsilon_B/2D_{max}}+1}$, the sampled perturbation over graph matrix satisfies $\epsilon_B$-DP.*

The proof is shown in Appendix A.2.2. For label perturbation, node $v_i$'s label $y_i$ is encoded via Random Response for it outperforms other oracles in low dimensions. The transformation is:

$$
p(y'_i|y_i) = \begin{cases} \dfrac{e^{\epsilon_C}}{e^{\epsilon_C} + M - 1}, \text{if } y'_i = y_i \\ \dfrac{1}{e^{\epsilon_C} + M - 1}, \text{ otherwise,} \end{cases} \tag{13}
$$

where $\epsilon_C$ is the privacy budget, and M is the class number. We obtain the differentially private label matrix $\tilde{\mathbf{Y}}$.

For the adaptive calibrated aggregation, a node-wise adaptive embedding aggregation and the residual connection cooperate to balance the smoothness of different perturbed embeddings between neighbors and the node itself. The process includes two steps: neighbor aggregation, and residual combination. First, for $k$-th layer, ACA aggregates neighbors equally based on the normalized perturbed adjacent matrix and previous layers outputs as $\mathbf{M}_u^{k-1} = \hat{\mathbf{A}}\mathbf{H}_u^{k-1}$, where $\hat{\mathbf{A}} = \hat{\mathbf{D}}^{-\frac{1}{2}}(\overline{\mathbf{A}'} + \mathbf{I})\hat{\mathbf{D}}^{-\frac{1}{2}}$, $\overline{\mathbf{A}'} + \mathbf{I}$ is the sampled noisy adjacent matrix with self-loop, $\mathbf{D}$ is the degree matrix, $H_u^0 = \overline{H_u^0}$. Then, for the residual combination, the intuition is to use the node's initial noisy embedding as a baseline. Larger changes in neighbor features suggest more irrelevance noise, thus their influence should be reduced, and vice versa. Specially, the residual weight $\gamma_u$ for $v_u$ is computed as:

$$
\gamma_u = \max(1 - \tau \cdot \frac{1}{\text{Dist}(\mathbf{M}_u^{k-1} - \overline{\mathbf{H}_u^0})}, 0), \tag{14}
$$

where $\tau$ is a learnable parameter that controls the smoothing, $\max(\cdot)$ is used to ensure weight is not smaller than zero, $\text{Dist}(\cdot)$ is the distance measure method. We use the widely used euclidean norm. Finally, the residual aggregation output of $v_u$ is the cooperation of $\overline{\mathbf{H}_u^0}$ and $\mathbf{M}_u^k$ weighted by $\beta_u$ as:

$$
\mathbf{H}_u^k = (1 - \gamma_u)\overline{\mathbf{H}_u^0} + \gamma_u \mathbf{M}_u^{k-1}. \tag{15}
$$

After $K$ hops adaptive aggregation, the final output of ACA is $\mathbf{H}_u^K$ for $\forall v_u \in \mathcal{V}$, which is used to the classification layer. Since both the generation of $\mathbf{H}^K$ and the perturbed label matrix $\tilde{\mathbf{Y}}$ satisfy DP guarantees, the training of the classification layer, which takes $\mathbf{H}^K$ and $\tilde{\mathbf{Y}}$ as inputs, does not require additional noise. Now we have theorem 5. The proof is shown in Appendix A.2.3.

**Theorem 5.** *With the aid of proposed private NAP and ACA, the whole training and inference process of our NIP-GNN preserves $(\epsilon_A + \epsilon_B + \epsilon_C)$-DP.*

## 5 EXPERIMENTAL EVALUATION

### 5.1 EXPERIMENT SETTINGS

**Datasets.** Four publicly available datasets are used, Cora, Citeseer (Yang et al., 2016), Lastfm (Rozemberczki & Sarkar, 2020), and Facebook (Rozemberczki et al., 2021). Cora and Citeseer are typical citation networks, representing two sparse graphs. Lastfm and Facebook are social networks, representing dense graphs. Detailed information is shown in Appendix A.3.

**Compared Methods.** We compare our proposed NIP-GNN with three state-of-the-art methods achieving node-level DP for GNN and one feature-based baseline method: 1) **DP-SAGE** Daigavane et al. (2021) is a one-layer GNN based on GraphSAGE Hamilton et al. (2017) that samples 1-hop neighbors and ensures DP by perturbing the gradient using DP-SGD. 2) **DPDGC** Chien et al. (2023) decouples the protection of topology and features by encoding them separately with DP-encoders on a bounded graph, then combines them using residual connections and trains with a DP-based classifier. 3) **GAP** Sajadmanesh et al. (2022) uniformly perturbs the aggregation on a bounded graph

and uses a DP-classifier to train on the perturbed embeddings. 5) **DP-MLP** is a feature-based baseline that only considers node features as input for training the private GNN, without incorporating any graph topology information. DP-MLP utilizes DP-SGD, ensuring DP by perturbing the gradients. This baseline is included to analyze whether noise affects the benefit of using graph structural information in GNN training. Our NIP-GNN is referred to as **Ours** in the following sections.

**Training and Evaluation Settings.** Following (Sajadmanesh et al., 2022), we split the nodes into a training set (75%), validation set (10%), and test set (15%) in a transductive setting. Since the original Lastfm is highly imbalanced, we choose the top 10 classes that have the most samples (Sajadmanesh & Gatica-Perez, 2021). Notably, unlike Sajadmanesh et al. (2022); Sajadmanesh & Gatica-Perez (2023); Chien et al. (2023), we do not preprocess the graph to bound the maximum degree to prevent information loss from edges and nodes. We use the same 2-layer neural network with a hidden embedding size of 64 as in Sajadmanesh et al. (2022). The training epochs are set as 100 with batch sizes of 64 across all datasets. All models are trained based on Adam optimizer (Kingma & Ba, 2014). The initial value of the learning rate is 1e-3, and the decay mechanism is used with a patience of 20 and a decay rate of 0.5. We measure the model performance by training 10 consecutive rounds on the test set and taking the average value with a 95% confidence interval under bootstrapping with 2000 samples. All experiments are implemented by using Pytorch and PyTorch-Geometric (PyG).

## 5.2 RESULTS AND ANALYSIS

### 5.2.1 EVALUATION ON MODEL UTILITY OVER DIFFERENT $\epsilon$

Table 1: Utility (Mean Accuracy $\pm$ 95% Confidence Intervals) comparisons with baselines on each graph datasets under $\epsilon$=4. The best performing private method is **highlighted**.

| Dataset | DP-MLP | DP-SAGE | DPDGC | GAP | NIP-GNN |
|---------|--------|---------|-------|-----|---------|
| Cora | 46.35$\pm$1.74 | 13.54$\pm$2.33 | 40.71$\pm$2.35 | 31.50$\pm$0.72 | **53.21$\pm$1.16** |
| Citeseer | 21.41$\pm$1.52 | 19.94$\pm$4.82 | 36.52$\pm$1.26 | 31.60$\pm$4.63 | **48.33$\pm$2.02** |
| Lastfm | 25.27$\pm$2.37 | 26.72$\pm$1.74 | 41.25$\pm$1.63 | 37.27$\pm$4.63 | **48.33$\pm$2.02** |
| Facebook | 48.35$\pm$1.74 | 33.52$\pm$2.62 | 37.81$\pm$2.35 | 43.50$\pm$0.73 | **51.11$\pm$1.48** |

**Compare with existing private GNNs.** We first compare the utility between the NIP-GNN and competitors under $\epsilon$=4. The results are shown in Table 1. Similar to previous studies Sajadmanesh et al. (2022); Sajadmanesh & Gatica-Perez (2023); Chien et al. (2023), the widely-used metric test accuracy is used to quantify the model utility. Ours achieves better utility under the same privacy level compared to GAP, DPDGC, DP-SAGE, and DP-MLP. These demonstrate the effectiveness of our NIP-GNN in balancing privacy and utility under the same level of DP guarantee.

### 5.2.2 ABLATION STUDIES OF PROPOSED MODULES

To study the impact of adaptive privacy budget allocation, degree-preserving sampling, and adaptive residual aggregation on model utility, we compared the effects of removing adaptive privacy budget allocation and replacing it with uniform allocation (**Ours w/o A-a**), removing degree-preserving sampling (**Ours w/o D-s**), and removing adaptive residual aggregation and replacing into SAGE (**Ours w/o A-r**) in Table 2 and 3. The experiments were conducted under two privacy budget settings ($\epsilon = 7$ and $\epsilon = 12$). Removing either proposed module leads to a decrease in utility, indicating that proposed adaptive privacy budget allocation, degree-preserving sampling, and adaptive residual aggregation significantly enhance utility.

### 5.2.3 EVALUATION ON RESILIENCE AGAINST MEMBERSHIP INFERENCE ATTACK

We empirically measure the privacy guarantee of Ours and other node-level private baselines and Ours w/o A-a (removing adaptive privacy budget allocation and replacing into uniform allocation) by conducting **node-level Member Inference Attack (MIA)**. We follow the TSTF approach (train on subgraph and test on full graph) Olatunji et al. (2021) as MIA and use the same architecture and settings as the target model. We use accuracy as the metric as the attack goal can be modeled as a balanced binary problem: whether the arbitrary node exists in the training graph. Table 4 reports

Table 2: Ablation studies under $\epsilon$=7.

| Dataset | Cora | Citeseer | Lastfm | Facebook |
|---|---|---|---|---|
| **Ours w/o A-r** | 44.3±0.36 | 31.5±0.48 | 51.6±0.69 | 50.1±0.91 |
| **Ours w/o A-a** | 63.4±0.26 | 42.4±0.17 | 65.3±0.15 | 65.5±1.10 |
| **Ours w/o D-s** | 64.5±1.30 | 43.9±2.07 | 64.9±0.14 | 67.1±0.09 |
| **Ours** | **64.9±1.75** | **44.6±0.26** | **65.9±0.15** | **67.8±0.17** |

Table 3: Ablation studies under $\epsilon$=12.

| Dataset | Cora | Citeseer | Lastfm | Facebook |
|---|---|---|---|---|
| **Ours w/o A-r** | 67.2±0.27 | 58.9±0.50 | 73.3±0.50 | 86.0±1.44 |
| **Ours w/o A-a** | 83.1±0.11 | 64.5±0.21 | 84.5±0.10 | 89.8±0.07 |
| **Ours w/o D-s** | 81.5±0.15 | 63.8±1.14 | 85.4±1.87 | 90.0±1.07 |
| **Ours** | **83.5±1.84** | **65.0±0.16** | **86.3±0.84** | **91.3±0.64** |

the mean accuracy with 95% confidence intervals of MIA attacks under four different privacy levels on all datasets. For private models, we can see that private GNNs can effectively defend against the attack, reducing the accuracy to around 50%, which is a nearly random selection. Ours and baselines have similar noise resilience across different privacy budgets, while Ours achieves a better trade-off between privacy and utility as shown in Table 1 before. The resilience gap between Ours and Ours w/o A-a shows the priority of the proposed adaptive budget scheme in the privacy leakage defense.

Table 4: Membership Inference Attack accuracy (Mean ± 95% Confidence Intervals) comparisons with baselines on each graph datasets under $\epsilon$=4.

| Dataset | DP-MLP | DP-SAGE | DPDGC | GAP | Ours w/o A-a | Ours |
|---|---|---|---|---|---|---|
| Cora | 49.78±3.03 | 50.09±1.06 | 49.58±1.44 | 50.12±2.17 | 50.21±1.91 | 49.95±1.30 |
| Citeseer | 51.18±0.77 | 49.15±1.45 | 50.76±0.57 | 48.04±1.41 | 49.98±1.32 | 49.01±0.25 |
| Lastfm | 50.67±0.57 | 49.42±0.36 | 49.50±0.89 | 48.99±0.69 | 50.25±1.39 | 48.23±0.33 |
| Facebook | 51.32±1.26 | 50.09±2.03 | 49.58±1.44 | 50.12±2.17 | 49.18±1.14 | 48.01±1.30 |

### 5.2.4 EVALUATION ON ADAPTIVE BUDGET OVER DIFFERENT $\epsilon$

Under the same global privacy budget, we also compare different noise scale ratios between differential private aggregation layer $\mathcal{H}^0$, the degree-preserving adjacent matrix $\tilde{\mathbf{A}}$ and noised label $\tilde{\mathbf{Y}}$. The result is shown in Figure 1. With noise on $\tilde{\mathbf{A}}$ be constant, we add more noise on $\mathcal{H}^0$ (decline $\epsilon_A$). Compared with the equal distribution method, the accuracy improvement represents adaptive DP on $\mathcal{H}^0$ mitigates the noise waste problem caused by node uniform privacy distribution. The scheme that assigns less $\epsilon$ on $\mathcal{H}^0$ converges faster. As $\epsilon$ increases (noise decreases), the model's utility tends to be the same for the two compared schemes.

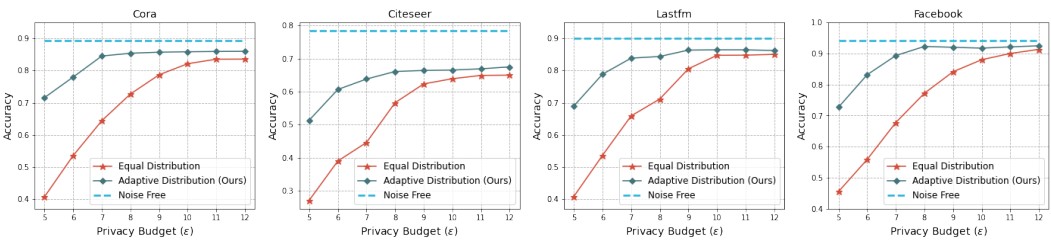

Figure 1: Evaluation on Different Privacy Budget.

### 5.2.5 EVALUATION ON MAXIMUM DEGREE $D_{max}$ OF GRAPH

We analyze the impact of the maximum node degree of the graph on model performance. The performance variation of the adaptive differential privacy mechanism is shown in Figure 2(a)-(d). The accuracy of the model increases to a peak and then decreases as $D_{max}$ increases. The maximum degree of the sub-graphs taken at the peak differs on data with different degree average values. When $D_{max}$ increases, the neighbor information that nodes can aggregate grows, but at the same time, more noise is received, thus affecting the accuracy of the model. Meanwhile, the Laplace mechanism is affected by data sensitivity. When $D_{max}$ increases, the data sensitivity increases, and thus the noise added to each node on average increases, reducing the utility of the model.

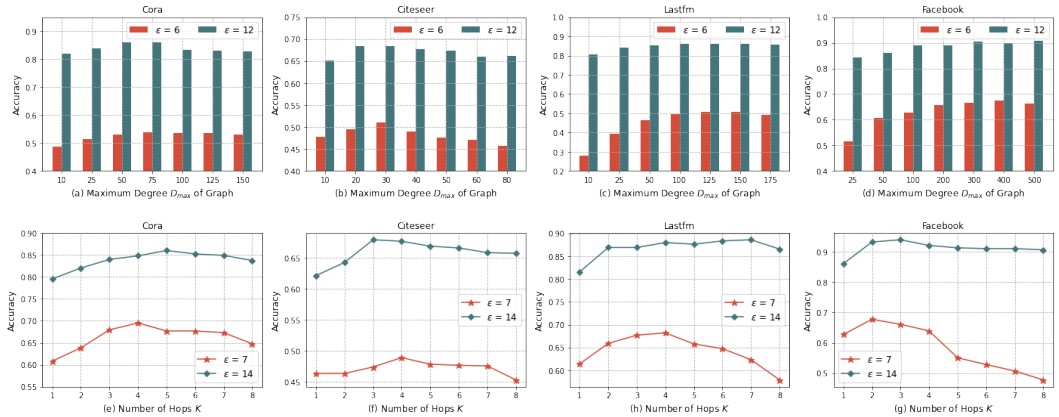

Figure 2: Evaluation on Maximum Degree $D_{max}$ of Graph (a-d) and Hop $K$ of ACA (e-g).

### 5.2.6 EVALUATION ON MULTI-ADAPTIVE-LAYER $K$

We investigate the effect of different hops on the model performance. The results are shown in Figure 2(e)-(g). As can be seen, both NIP-GNN and its competitors can aggregate more information from neighbors when $K$ increases. However, there is a trade-off between $K$ and accuracy. When the value of $K$ increases, the accuracy of both NIP-GNN and its competitors increases first, then reaches a peak and decreases. This is because when more layers of neighbor information are aggregated, the noise data collected from the neighbors also increases, affecting the behavior of the model. Besides, when $\epsilon$ is small, the noise scale of the joined data is larger, and the model needs higher $K$ to reach the peak.

## 6 CONCLUSION

In this work, we proposed a Node-level Individual Private Graph Neural Network (NIP-GNN) that flexibly adjusts node protection levels based on learnable influence, independent of the training epoch, to address the privacy-utility trade-off in GNNs. We introduced the Topology-based Node Impact Estimation (TNIE) method to capture node influence through neighborhood and centrality awareness. Additionally, we developed an influence-aware fine-grained permutation method that injects diversity-calibrated noise to achieve node-level privacy-preserving distributions. To further mitigate the negative impact of noise on model utility, we designed an adaptive message-passing layer with residual selective cooperation. Our theoretical analysis confirms that NIP-GNN satisfies differential privacy guarantees, and experimental results on real-world graph datasets demonstrate that our approach achieves a better utility-privacy trade-off compared to existing private GNN methods. Notably, this work is the first to demonstrate the applicability of differentiated noise addition in GNNs, providing a new direction for future research in privacy-preserving graph learning.

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

# A  APPENDIX

## A.1  OTHER RELATED WORKS

### A.1.1  OTHER DIFFERENTIALLY PRIVATE GNNS

The Locally Private Graph Neural Network (LPGNN) Sajadmanesh & Gatica-Perez (2021) realizes the GNN model framework under Local Differential Privacy (LDP) guarantee in a decentralized setting. Solitude Lin et al. (2022) preserves edge-level and node-level simultaneously under LDP by calibrating the added noise on the graph. Another edge-level differential privacy GNN algorithm was proposed in Wu et al. (2022) by perturbing the adjacency matrix of the graph through the Laplace mechanism. Blink in Zhu et al. (2023) provides edge-level LDP via spending the privacy budget separately on links and degrees of the graph and denoising the topology using Bayesian estimation. PrivGNN Olatunji et al. (2023b) provides node-level DP by adapting the framework of PATE Papernot et al. (2016). The student GNN model is trained using public graph data, where each node is privately labeled by teacher GNN models. These teacher models are exclusively trained for their respective query nodes. However, PrivGNN is dependent on the availability of public graph data. DPDGC Chien et al. (2023) proposed a unified notation, Graph Differential Privacy (GDP), in GNN. It points out that topology and nodes may need multi-granular protection.

### A.1.2  PRIVACY ATTACKS IN GNNS

Several studies have explored the privacy leakage in GNNs, involving membership inference attacks Wang & Wang (2022); Zhang et al. (2022b), attribute inference attacks Olatunji et al. (2023a), model stealing attacks Shen et al. (2022), edge stealing attacks Wu et al. (2022); He et al. (2021) and graph reconstruction attacks Zhang et al. (2022b). Authors in Wang & Wang (2022) infer the node & link group distribution under both white-box and black-box settings. Attribute inference attacks aim to infer the node attribute of the graph with only access to GNN output or embedding. The Study in Olatunji et al. (2023a) shows that even under the black-box setting, the attacker is able to infer the sensitive attribute by some public attributes and graph structure. Work in Zhang et al. (2022b) aims to infer the basic group properties like node and edge number, and graph density and determine whether the subgraph exists in the original graph and graph reconstruction attacks with only black-box access. A study in Shen et al. (2022) shows that pure GNNs face model stealing risks. Edge stealing attack aims to infer whether there exists a link between any pair of nodes with only access to GNN outputs. Experiment results in Wu et al. (2022); He et al. (2021) show that the existence of edges is vulnerable due to graph attributes like heterogeneous.

### A.1.3  PRIVACY BUDGET ALLOCATION

For traditional differentially private DL methods, several works have proposed privacy budget allocation strategies to address specific scenarios. It can be broadly categorized into three types: Preference-based Allocation, Sensitivity-based Allocation, and Utility-based Allocation. In preference-based allocation, Li et al. (2017) introduces a sensitivity-based differential privacy budget partitioning mechanism to meet the different privacy preferences of data owners. Boenisch et al. (2024) focuses on grouping and allocating according to each user's privacy needs, proposing a demand-based sampling and scaling mechanism that ensures data points with higher privacy budgets are sampled more frequently during training and adjusts the scale of noise accordingly. In utility-based allocation, Niu et al. (2020) proposes the Utility-aware Personalized Exponential Mechanism (UPEM), which considers the quantitative changes of the Personalized Exponential Mechanism to enhance the overall utility of the model. Feldman & Zrnic (2021) introduces individualized privacy accounting, where privacy budgets are allocated based on the privacy loss of each data point during training. In risk-based allocation, Jorgensen et al. (2015) proposes a personalized differential privacy budget allocation method based on individualized risk levels.

### A.2 Theoretical Analysis

#### A.2.1 Privacy Analysis of NAP over Sum Aggregation

**Lemma 1.** *Let $G = \{\mathcal{V}, \mathcal{E}\}$ and $G' = \{\mathcal{V}', \mathcal{E}'\}$ be two adjacent graph. The global $L_1$ graph sensitivity of first sum aggregation layer $\Delta\mathcal{H}^0 \leq 2D_{max}$, where $D_{max}$ is the maximum node degree.*

*Proof.* Assume adjacent graph datasets $G$ and $G'$ differ in node $v_k$. Then we have:

$$\Delta\mathcal{H}^0 = \max_{n_u \in \mathcal{V}} || \sum \mathcal{H}^0(\mathbf{A}, \mathcal{V}, \mathbf{X}) - \sum \mathcal{H}^0(\mathbf{A}', \mathcal{V}', \mathbf{X}')||_1 = \max \sum_{v_u \in \mathcal{V}} || \sum_{j=1}^{|N|} (\mathbf{A}_{uj}\mathbf{X}_j - \mathbf{A}'_{uj}\mathbf{X}'_j)||_1 \tag{16}$$

Without loss of generality, in $G'$, we assume that node $v_k$ is removed from $G$. Therefore, for nodes $v_i$ and $v_j$, we have $\mathbf{A}'_{ij} = 0$, if $i = k$ or $j = k$, otherwise $\mathbf{A}'_{ij} = \mathbf{A}_{ij}$. Then we have:

$$\Delta\mathcal{H}^0 = || \sum_{j=1}^{|N|} \mathbf{A}_{kj}\mathbf{X}_j + \sum_{i=1}^{|N|} \mathbf{A}_{ik}\mathbf{X}_k ||_1 \leq \sum_{j=1}^{|N|} \mathbf{A}_{kj} + \sum_{i=1}^{|N|} \mathbf{A}_{ik} \leq D_k + D_k \leq 2D_{max}, \tag{17}$$

where $D_k$ is the degree of $v_k$. The lemma is proved. $\qquad\square$

**Theorem 3.** *Algorithm 1 preserves $\epsilon_A$-DP in the first differential private aggregation layer $\mathcal{H}^0$.*

*Proof.* All nodes' aggregation embedding in $G$ are perturbed, therefore we have:

$$Pr(\overline{\mathcal{H}^0}(\mathbf{A}, \mathcal{V}, \mathbf{X})) = \prod_{i=1}^{N} exp(\frac{\epsilon_i}{\Delta\mathcal{H}^0}|| \sum_{j=1}^{N} \mathbf{A}_{ij}\mathbf{X}_j j - \overline{\mathbf{H}_i^0}||_1). \tag{18}$$

$\Delta\mathcal{H}^0$ is set to $2D_{max}$, as proved in Lemma 1. Assume adjacent graph datasets $G$ and $G'$ differ in node $v_k$. We have:

$$\frac{Pr(\overline{\mathcal{H}^0}(\mathbf{A}, \mathcal{V}, \mathbf{X}))}{Pr(\overline{\mathcal{H}^0}(\mathbf{A}', \mathcal{V}', \mathbf{X}'))} \leq \prod_{i=1}^{N} exp(\frac{\epsilon_i}{\Delta\mathcal{H}^0}|| \sum_{j=1}^{N} \mathbf{A}_{ij}\mathbf{X}_j j - \sum_{j=1}^{N} \mathbf{A}'_{ij}\mathbf{X}'_j ||_1)$$

$$= exp(\sum_{i=1}^{N}\sum_{j=1}^{N} \frac{\epsilon_i}{\Delta\mathcal{H}^0}|| \mathbf{A}_{ij}\mathbf{X}_j - \mathbf{A}'_{ij}\mathbf{X}'_j ||_1) = exp(\sum_{v_j \in N(k)} \mathbf{A}_{kj}\frac{\epsilon_k}{\Delta\mathcal{H}^0} + \sum_{v_i \in N(k)} \mathbf{A}_{ik}\frac{\epsilon_i}{\Delta\mathcal{H}^0}). \tag{19}$$

Let $f(k) = \sum_{v_j \in N(k)} \mathbf{A}_{kj}\epsilon_k + \sum_{v_i \in N(k)} \mathbf{A}_{ik}\epsilon_i$. Then we have:

$$f(k) = D_k\epsilon_k + \sum_{v_i \in N(k)} \epsilon_i = \epsilon_A(D_k\beta_k + \sum_{v_i \in N(k)} \beta_i)$$

$$\leq \epsilon_A(D_k\beta_k + \sum_{v_i \in N(k)} (\frac{\mathcal{R}\_\mathcal{NI}(u)}{\sum_{v_j \in N(k)} \mathcal{R}\_\mathcal{NI}(j)}(D_{max} - D_k + 1))) \tag{20}$$

$$\leq \epsilon_A(D_k\frac{D_{max}}{D_k} + D_{max} - D_k + D_k) \leq 2\epsilon_A D_{max}.$$

Substitute equation (20) into (19), we can get:

$$\frac{Pr(\overline{\mathcal{H}^0}(\mathbf{A}, \mathcal{V}, \mathbf{X}))}{Pr(\overline{\mathcal{H}^0}(\mathbf{A}', \mathcal{V}', \mathbf{X}'))} \leq exp(\frac{2\epsilon_A D_{max}}{\Delta\mathcal{H}^0}) = exp(\epsilon_A). \tag{21}$$

$\qquad\square$

**Proposition 2.** *The sum aggregator function defined in (9) for the first layer is unbiased.*

*Proof.* The expectation of noised embedding for node $v_u$ is:

$$\mathbb{E}[\overline{\mathbf{H}_u^0}] = \mathbb{E}[\sum_{j=1}^{|N|} \mathbf{A}_{uj}\mathbf{X}_j + Lap(\frac{\Delta\mathcal{H}^0}{\epsilon_u})] = \mathbb{E}[\sum_{j=1}^{|N|} \mathbf{A}_{uj}\mathbf{X}_j] + \mathbb{E}[Lap(\frac{\Delta\mathcal{H}^0}{\epsilon_u})] = \mathbb{E}[\mathbf{H}_u^0]. \tag{22}$$

Proposition 2 is proved. $\qquad\square$

### A.2.2 PRIVACY ANALYSIS OF EDGE PERTURBATION

**Theorem 4.** *Let $D_u$ denote the degree of $v_u$ in the original matrix $\mathbf{A}$, $\overline{D'_u}$ denotes the degree after perturbation and sampling. Then $\mathbb{E}(\overline{D'_u}) = \mathbb{E}(D_u)$. When the privacy parameter $s$ satisfies $s \geq \frac{2}{e^{\epsilon_B/2D_{max}}+1}$, the sampled perturbation over graph matrix satisfies $\epsilon_B$-DP.*

*Proof.* In the original adjacent matrix $\mathbf{A}$, there are $D_u$ 1s and $(N - D_u)$ 0s for node $v_u$. The expectation of $v_u$'s degree after edge perturbation is:

$$\begin{aligned} \mathbb{E}(\overline{D_u}) &= D_u s + D_u(1-s) + 0.5(N - D_u)(1-s) \\ &= 0.5D_u + 0.5N - 0.5Ns + 0.5D_u s, \end{aligned} \tag{23}$$

where $s$ is the Bernoulli sample probability. Then we have:

$$\mathbb{E}(\overline{D'_u}) = \mathbb{E}(\overline{D_u}) * p_u^{sample} = \mathbb{E}(D_u). \tag{24}$$

Assume adjacent graph $G$ and $G'$ differ in node $v_k$. Denote the edge perturbation function as $\mathcal{P}$, then we have:

$$\frac{\mathcal{P}(\mathbf{A})}{\mathcal{P}(\mathbf{A}')} = \prod_{v_i \in N(k)} \frac{\mathcal{P}(\mathbf{A}_{ik})}{\mathcal{P}(\mathbf{A}'_{ik})} \frac{\mathcal{P}(\mathbf{A}_{ki})}{\mathcal{P}(\mathbf{A}'_{ki})} = (\frac{1-s/2}{s/2})^{D_k+D_k} \leq (\frac{1-s/2}{s/2})^{2D_{max}}. \tag{25}$$

When $s \geq \frac{2}{e^{\epsilon_B/2D_{max}}+1}$, we have $\mathcal{P}(\mathbf{A})/\mathcal{P}(\mathbf{A}') \leq exp(\epsilon_B)$. The theorem is proved. $\square$

### A.2.3 PRIVATE ANALYSIS OF WHOLE NIP-GNN

**Theorem 5.** *With the aid of the proposed private NAP and ACA, the whole training and inference process of our NIP-GNN preserves $(\epsilon_A + \epsilon_B + \epsilon_C)$-DP.*

*Proof.* From theorem 3 and 4, we know that the first aggregation layer satisfies $\epsilon_A$-DP and the sampled adjacent matrix perturbation guarantees $\epsilon_B$-DP. Random response mechanism on node label guarantees $\epsilon_C$-DP under equation (13). Node-level DP preserves all information of one node, including features, edges, and labels. Differential aggregation layer $\mathcal{H}^0$ process node feature and edge privately. The following ACA does not expose node features and edges for it only post-processing the noised aggregation embedding and perturbed adjacent matrix without access to private features and links. Therefore, following the post-processing and sequential composition theorem (refer to Theorem 1 and 2), the training and inference phase also guarantees DP because node label is perturbed former, ensuring that every sensitivity component (node features, edges, labels) is protected. The total privacy cost is $(\epsilon_A + \epsilon_B + \epsilon_C)$-DP. $\square$

### A.3 DETAILS OF USED DATASETS

Detailed information on experimental datasets is shown in Table 5.

Table 5: Detailed Statistic of Used Datasets

| Datasets | Nodes | Edges | Features | Avg_Deg |
|---|---|---|---|---|
| Cora | 2708 | 5278 | 1433 | 3.89 |
| Citeseer | 3327 | 4552 | 3703 | 3.73 |
| Lastfm | 7083 | 25814 | 7842 | 8.28 |
| Facebook | 22470 | 170912 | 4717 | 15.21 |

