# OpenReview forum: "Individualized Private Graph Neural Network via Node Influence-based Noise Adaptation"
_ICLR.cc/2025/Conference — Submitted to ICLR 2025_

### Official Review · Reviewer_5h8v · 2024-10-26

**Soundness:** 3
**Presentation:** 3
**Contribution:** 2
**Rating:** 6
**Confidence:** 3

**Summary:**

The author proposed NIP-GNN that is aimed at the lack of the flexibility to adjust between nodes. They propose a Node-level Individual Private GNN with DP guarantee based on the adaptive perturbation over sensitive components to safeguard node information.

**Strengths:**

The idea is interesting, and the theoretical proof is complete.

**Weaknesses:**

1. You say that your model could get a trade-off between the privacy and performance, how did you show this in the experiment?

**Questions:**

1. What does the shifting degree mean? You could explain it more clearly.
2. The score function seems to like the aggregation of GCN. You should explain why you choose to use GCN rather than GIN or GraphSage?

---

### Official Review · Reviewer_TXeS · 2024-10-26

**Soundness:** 2
**Presentation:** 1
**Contribution:** 2
**Rating:** 3
**Confidence:** 4

**Summary:**

The authors study the problem of differentially private graph neural networks. They aim to leverage the influence of each node to adaptively control the noise allocation. They show the effectiveness of their approach via experiments.

**Strengths:**

- The studied problem of DP-GNN is important.
- The paper is well-written up to Section 3.1.

**Weaknesses:**

- The definition of privacy pertaining to graph settings is unclear.
- The clarity of Section 4 needs significant improvement.
- The logic and reason of the proposed methods are not conveyed clearly.

## Detail comments

I have mixed feelings when reading this paper. Everything before Section 3.1 is very well-written, but Section 4 (methodology) makes me feel like it is written by a completely different person. It is not polished, and there are many typos and errors in the notations. The excessive amount of those errors even makes it hard to understand why the proposed method is reason at a high level. It is clearly subpar of top ML conferences like ICLR and I hope the authors can at least polish their Section 4 to the quality of their introduction. Please see my question sections for the subset of errors and typos that I found.

Another issue is that the privacy definition in Section 3.2 is not clear. Is Definition 1 based on add-remove DP or replacement DP? If it is the former as in Sajamanesh et al., 2022, then the corresponding definition 4 is problematic as explained in Chien et al. 2023. Note that removing one node can already cause the dimension of the node embedding matrix *$H$* to be different on the adjacent dataset, which makes it impossible to achieve DP. It might make sense if the authors adopt the replacement DP definition, but in this case, the required noise is also large as argued in Chien et al. 2023. I feel the authors should have a detailed discussion on why exactly the privacy definition they are considering.

In summary, I really like the idea that the authors proposed. I do think allocating privacy budget/noise according to the noise importance/influence is the correct direction for better privacy-utility trade-off in DP-GNN. Unfortunately, the authors did not convey clearly why the approach they proposed makes sense. I hope the authors can carefully polish their manuscript and really make this great idea work.

**Questions:**

1.	What is *$u$* in equation (6)? It seems never defined.
2.	Also, what exactly does the subscript $j$ mean? It is also not clearly explained.
3.	Why call equation (7) neighborhood sensing? Isn’t it just the normal graph propagation/neighborhood aggregation? I do not see the reason why we need to diverge from the terminology of the standard graph learning literature.
4.	What loss is used to train an MLP for knowing the feature importance? What exactly is the Score computing function? Is it a loss function or just the MLP? Why such a design can achieve feature-awareness mining?
5.	In equation (5), is $F(G)$ the output of GNNs or its weight? If it is the output, does it contain the predicted labels of ***all*** nodes? This will cause the issue that I mentioned in the weaknesses section.
6.	How do you train the learnable parameters in equation (8)? How do you ensure the DP properties there?
7.	In equation (9), the subscript of X is off.
8.	Line 275, the $*$ should be the superscript of $s(u)$?
9.	Why Laplace mechanism with $\ell_1$ sensitivity, but not Gaussian mechanism with $\ell_2$ sensitivity as in prior works?

---

### Official Review · Reviewer_Z7iH · 2024-11-02

**Soundness:** 2
**Presentation:** 1
**Contribution:** 2
**Rating:** 5
**Confidence:** 4

**Summary:**

The authors propose NIP-GNN, a privacy-preserving GNN framework that aims to address the limitations of uniform noise mechanisms in existing Differential Privacy (DP) GNNs. NIP-GNN introduces a noise scheme based on a node's estimated influence, leveraging a Topology-based Node Influence Estimation (TNIE) method. This approach applies fine-grained noise to protect nodes while theoretically meeting DP guarantees. The paper evaluates NIP-GNN on benchmark datasets and claims improved utility-privacy trade-offs and resistance to inference attacks.

**Strengths:**

The approach that calibrates noise based on node influence addresses an issue in current DP-GNNs, where uniform noise injection can disproportionately impact node utility. By tailoring noise to node influence, NIP-GNN reduces unnecessary noise for low-influence nodes, aiming to improve utility without compromising privacy.

**Weaknesses:**

1.	The research problem is suggested to be formally proposed in section 3.2.
2.	Wrong statement. Note that the community recognizes that a lower epsilon indicates a higher level of privacy protection. However, the statement in line 178 violates the definition of DP. The “Compared” should be “compared” in line 308. Authors are suggested to proofread and revise this paper carefully.
3.	Vague statement. Authors are suggested to explain the motivation of section 4.1, the reusable privacy ratio in line 286, and the process of obtaining the budget ratio of equation 11. The derivation of equations 12 and 14 is not provided. In the appendix (lin 772), the first + is obtained without showing details. Authors are suggested to carefully revise this paper by providing more details and a smooth logical flow.
4.	Missing evaluations. According to definition 1 of this paper, the neighbour graphs are different in their node features, links and labels. This paper also involves the design of edge and label noises. However, this evaluation only focuses on the MIA and ignores the evaluation of edge privacy and label privacy. Authors are suggested to provide sufficient privacy evaluation to verify the effectiveness of the proposed method. A possible evaluation method could be found in Meng, Lingshuo, et al. "Devil in Disguise: Breaching Graph Neural Networks Privacy through Infiltration." Proceedings of the 2023 ACM SIGSAC Conference on Computer and Communications Security. 2023.
5.	Limited Scalability Evaluation and Efficiency Concerns: The adaptive noise allocation relies on calculating each node's influence based on the centrality and neighbourhood structure, which is computationally expensive, especially for large or dense graphs. A scalability analysis in terms of time complexity and memory usage would strengthen the paper’s claims on practicality and applicability to real-world datasets. Consider adding a complexity analysis and reporting runtime and memory usage across different graph sizes and densities. Alternatively, discuss possible optimizations to the TNIE calculation, such as approximations for influence estimation that reduce computational cost without sacrificing significant accuracy. Exploring parallel processing or sampling methods for large graphs could also enhance scalability.

**Questions:**

Refer to the weakness

---

### Official Review · Reviewer_rrVf · 2024-11-03

**Soundness:** 2
**Presentation:** 2
**Contribution:** 2
**Rating:** 3
**Confidence:** 4

**Summary:**

This paper presents a node-level private GNN with DP guarantee. It estimates each node’s influence and adds noise according to the influence rank.

**Strengths:**

The proposed solution is able to adjust the injected noise according to the node degree.

**Weaknesses:**

1. The literature review and experimental comparisons are insufficient. At least two relevant works on node differential privacy (DP) in GNNs, DPAR (https://arxiv.org/abs/2210.04442v3) and https://arxiv.org/abs/2311.06888, are omitted. Experimental results on the Cora (similar to Cora-ML used in DPAR) and Facebook datasets show that this paper’s performance is significantly inferior to these papers. To provide a fair comparison, the authors should benchmark all methods on Cora-ML, Pubmed, and Facebook datasets, using multiple values of epsilon rather than just one, as currently done.

2. Definition 2 incorrectly states that a "higher epsilon means higher protection," when in fact a higher epsilon implies weaker privacy protection.

3. There is no explanation for the vector “u” in Equation (6), leaving the reader unclear about its role.

4. Equations (7), (8), and (12) iteratively use node degrees to compute node embeddings and scores. However, node degrees vary between neighboring datasets. The paper lacks a DP mechanism to protect this private information and violates node privacy.

5. Theorem 3 references “Algorithm 1,” which does not appear in the paper. The paper would greatly benefit from presenting a complete algorithm to clarify each module’s execution order and purpose. It is unclear if the missing Algorithm 1 was intended to serve this function.

6. In this paper, noise is added to the aggregation results in the first layer of the GNN and then added again to the adjacency matrix used in subsequent layers. The rationale for this is unclear. Why not simplify it by directly adding noise to the adjacency matrix?

7. The experimental setup lacks a sufficient range of epsilon values. For utility experiments, epsilon should range from 4 to 16, as in https://arxiv.org/abs/2311.06888, to provide a more comprehensive evaluation.

8. Typos, e.g., in Section 4.3 and Eq. (20).

**Questions:**

N/A

---

### Meta-Review · Area_Chair_uHbA · 2024-12-14

**Metareview:**

The paper proposes to inject noise according to the node degree to enhance privacy.

All three reviewers and I, the AC, were unanimous in our assessment that this paper does not meet the bar of ICLR.

1) It is poorly written, and thus, hard to comprehend. Several notations used are not defined properly.
2) The experiments are not convincing - they lack scalability and efficiency considerations.
3) Due to the poor exposition, the method is clearly explained to readers. In fact, the definition of privacy pertaining to the graph setting is unclear.

**Additional Comments On Reviewer Discussion:**

There were no discussions.

---

### Decision · Program_Chairs · 2025-01-22

Reject